# Mortality and associated risk factors in patients with blood culture positive sepsis and acute kidney injury requiring continuous renal replacement therapy—A retrospective study

**Mikko J. Järvisalo**[1,2,3]*, **Tapio Hellman**[1], **Panu Uusalo**[2,3]

1 Kidney Center, Turku University Hospital and University of Turku, Turku, Finland, 2 Department of Anaesthesiology and Intensive Care, University of Turku, Turku, Finland, 3 Perioperative Services, Intensive Care and Pain Medicine, Turku University Hospital, Turku, Finland

* mikko.jarvisalo@tyks.fi

**Data Availability Statement:** The data underlying this study contain potentially identifying participant information and cannot be shared publicly. Future

## Abstract

### Objectives

Septic acute kidney injury (AKI) requiring continuous renal replacement therapy (CRRT) carries a mortality risk nearing 50%. Risk factors associated with mortality in AKI patients undergoing CRRT with blood culture positive sepsis remain unclear as sepsis has been defined according to consensus criteria in previous studies.

### Methods

Risk factors associated with intensive care unit (ICU), 90-day and overall mortality were studied in a retrospective cohort of 126 patients with blood culture positive sepsis and coincident severe AKI requiring CRRT. Comprehensive laboratory and clinical data were gathered at ICU admission and CRRT initiation.

### Results

38 different causative pathogens for sepsis and associated AKI were identified. ICU mortality was 30%, 90-day mortality 45% and one-year mortality 50%. Immunosuppression, history of heart failure, APACHE II and SAPS II scores, C-reactive protein and lactate at CRRT initiation were independently associated with mortality in multivariable Cox proportional hazards models. Blood lactate showed good predictive power for ICU mortality in receiver operating characteristic curve analyses with AUCs of 0.76 (95%CI 0.66–0.85) for lactate at ICU admission and 0.84 (95%CI 0.72–0.95) at CRRT initiation.

### Conclusions

Our study shows for the first time that lactate measured at CRRT initiation is predictive of ICU mortality and independently associated with overall mortality in patients with blood culture positive sepsis and AKI requiring CRRT. Microbial etiology for septic AKI requiring CRRT is diverse.

data access requests should be sent to the Ethics Committee of Southwest Finland Hospital District (eettinen.toimikunta@tyks.fi) or the Department of Anesthesiology and Intensive Care and the Informatics Department of Turku University Hospital via the corresponding author.

**Funding:** The author(s) received no specific funding for this work.

**Competing interests:** The authors have declared that no competing interests exist.

## Background

Sepsis is the predominant etiology of severe acute kidney injury (AKI) requiring renal replacement therapy (RRT) in critically ill patients [1]. The mechanisms underlying sepsis-associated AKI still, remain to be fully characterized [2]. Despite improvements in intensive care, mortality in sepsis patients receiving continuous RRT (CRRT) for AKI remains close to 50% within 90 days following intensive care admission [3,4]. Although, differences in methodologies employed for RRT at different intensive care units (ICUs) exist, CRRT is usually the modality of choice to treat critically ill AKI patients with the highest comorbidity and presence of hemodynamic instability.

Although a number of studies have explored risk factor associations with mortality in septic AKI, a large proportion of patients included in these studies have had a clinical diagnosis of sepsis defined according to the Sepsis-3 (suspected or confirmed infection with an increase $\geq 2$ in Sequential organ failure assessment score (SOFA)) [5] or earlier consensus criteria, and blood culture findings, whether positive or negative have not been described. Data on mortality and associated risk factors limited to patients with blood culture positive sepsis and AKI requiring CRRT are virtually non-existent. Furthermore, very limited data are available for blood stream infection pathogens leading to sepsis-associated AKI and the need for RRT.

It would be of great value to recognize risk factors associated with mortality in this patient group and assess individual mortality risk more precisely early on and potentially target treatments accordingly. Moreover, it would be important to characterize pathogens potentially leading to septic AKI requiring intensive care and CRRT.

Therefore, we aimed to study risk factors associated with ICU-, 90-day and overall mortality in blood culture positive sepsis and associated severe AKI requiring CRRT and to identify the responsible blood culture positive pathogens in this retrospective study.

## Methods

### Data sources, collection and study population

Patients admitted to the intensive care unit of Turku University Hospital, Turku, Finland, an academic tertiary medical center from January 1, 2010 through December 31, 2019, requiring CRRT and with ascertained diagnosis of sepsis and positive blood culture(s) were included in this retrospective study. Patients without sepsis (n = 229) and patients with sepsis but negative blood cultures (n = 115) were excluded, as were patients with maintenance dialysis dependency prior to ICU admission (n = 22). Blood culture findings had been examined case by case and considered to be relevant etiological agents for the sepsis present by an ICU infectious diseases senior consultant. In one patient with a single blood culture positive for *Staphylococcus capitis* the culture finding was considered a contamination, not a causative agent for sepsis by an infectious diseases specialist and a microbiologist and the patient was excluded from the study. Therefore, 126 patients were included in the analyses (S1 Fig). Every patient included in the study fulfilled the Sepsis-3 criteria for sepsis [5] in addition to presenting with blood culture positive bacteremia or fungemia. The individual patient data were collected from the hospital's medical documents with the permission of the Turku University Clinical Research Center scientific review board and the Hospital district of Southwest Finland. The data from the hospital software were combined and the patient identity numbers removed before the statistical analyses. For this retrospective, registry-based, non-interventional study the regulatory review board waived the need for informed consent in terms of data collection and analysis and publication of results.

For the purpose of this study, blood pH, bicarbonate, lactate, base excess, electrolytes, and other laboratory variables, blood pressure, need for invasive mechanical ventilation, PaO$_2$/

FiO$_2$-ratio, diuresis and vasopressors were recorded at ICU admission and at CRRT initiation. Other data extracted from patients' medical records included demographics, chronic medical conditions, fluid balance at CRRT initiation, CRRT dose and Acute Physiology and Chronic Health Evaluation (APACHE) II score, Simplified Acute Physiology Score (SAPS) II, and SOFA score. Estimated glomerular filtration rate (eGFR) was assessed at baseline (within one-year prior to ICU admission) and at 90 days and at one year after discharge in surviving patients.

## CRRT modality

Continuous Veno-Venous Hemodialysis for all patients was performed using Fresenius Multi-filtrate CRRT monitors and 1.80 m$^2$ polysulfone hemofilters Ultraflux AV1000 or Ultraflux EMiC2 HCO membranes with CiCa dialysate to achieve regional citrate anticoagulation (Fresenius Medical Care, Bad Hamburg, Germany). Post-filter-ionized calcium levels were used for anticoagulation monitoring. Blood and dialysate flow rates were prescribed according to the weight of the patient and by the caring ICU physician to target a dialysis dose of > 25 ml/kg/h. The methodology for CRRT remained unaltered for the entire study period.

## Statistical analysis

Results are presented as mean ± standard deviation (SD) for the normally distributed continuous variables and as median [inter-quartile range (IQR)] for skewed variables. Normality in continuous covariates was tested with Kolmogorov-Smirnov and Shapiro-Wilk tests. Student's t-test was used to compare continuous normally distributed covariates and Chi-square test for categorical covariates in the study subgroups. For skewed variables the comparisons between groups were done using a non-parametric Kruskall-Wallis test. Comparisons between eGFR at baseline and at 90 days and one year were done using paired t-tests. The relationship between mortality and exposure variables of interest was examined using univariable and multivariable Cox proportional hazard models. Variables that were significantly associated with mortality in univariable Cox models were included as covariates in stepwise multivariable Cox proportional hazards models. To avoid significant collinearity in the models in terms of laboratory and clinical data at ICU admission and CRRT initiation, multivariable analyses were performed by using two respective stepwise multivariable Cox models: one including significant baseline characteristics and laboratory variables at ICU admission and one with baseline characteristics and variables at CRRT initiation. Only variables with a significance P≤0.15 were included in the respective, final multivariable Cox models. Potential existence of multicollinearity was assessed by examining variance inflation factors. For the Cox models, all variables were further standardized to have a mean 0 and SD 1 in order to facilitate comparisons of the magnitudes of hazard ratios between the exposure variables. Receiver operating characteristics (ROC) curve analyses were conducted to estimate the area under the curve (AUC) as a measure of discriminative capacity of lactate at ICU admission and at CRRT initiation for mortality (ICU-mortality and 90-day mortality). Generally, we consider an AUC >0.90 outstanding, an AUC 0.80–0.90 excellent, an AUC 0.70–0.80 acceptable and an AUC <0.70 poor discrimination.

All statistical analyses were performed using statistical analysis system, SAS version 9.3 (SAS Institute Inc., Cary NC). P<0.05 was considered statistically significant.

## Statement of ethics

The study protocol (reference number J26/19) was approved by the Turku University Clinical Research Center scientific ethics review board and the Hospital district of Southwest Finland.

For this retrospective, register-based, non-interventional study the regulatory review board waived the need for informed consent in terms of data collection and analysis and publication of results.

## Results

### Patient characteristics

The demographic, clinical and laboratory data of the whole study cohort and comparisons between subgroups according to 90-day mortality are shown in Table 1 and the blood culture findings in Table 2. We identified 38 different pathogens in the blood cultures. All of the blood culture findings were examined case by case and considered to be relevant etiological agents for the sepsis present by the ICU infectious diseases senior consultant. Fifteen patients had a coinfection with at least two culture-positive pathogens in their blood samples. The most prevalent causative pathogens were: *Staphylococcus aureus* 25%, *Escherichia coli* 19%, *Streptococcus pneumoniae* 12%, *Streptococcus pyogenes* (A) 11% and *Candida albicans* 6%. Only one patient had septicemia caused by extended-spectrum beta-lactamase producing strain of *Escherichia coli*. Methicillin or carbapenem resistant strains of bacteria were not observed. The median time from ICU admission to diagnosis (positive blood culture(s)) was 0.3 (-0.6–1.3) days and the median time from the positive blood culture(s) to CRRT initiation was 0.5 (-0.6–1.7) days. The median number of blood culture samples per patient was 2 (2–4) and the median number of positive blood cultures per patient was 2 (2–2). The source of infection was blood-born in 56 (44%), skin/soft tissues in 35 (28%), urinary in 14 (11%), abdominal in 11 (9%) and respiratory in 10 (8%) patients, respectively. The antimicrobial therapies employed are shown in S1 Table. The antimicrobial treatment remained unchanged in 49 (39%) patients, in 28 (22%) patients an additional antimicrobial regimen was added to the treatment and in 49 (39%) patients the treatment was changed entirely after positive blood culture findings. The median time from the start of first empirical therapy to the initiation of specific antimicrobial therapy based on blood culture findings was 1 (1–2.5) days.

Mean SOFA score was 14.6±3.3, mean Simplified acute physiology II (SAPS-II) score 58.8 ±16.3 and mean Acute physiology and chronic health evaluation II (APACHE-II) score 26.5 ±7.2. Almost all of the study patients (95%) needed vasopressor support and 76% were on invasive mechanical ventilation during their ICU stay. A total of 90 patients (71%) were considered to have septic shock according to the Sepsis 3-criteria [5]. The median time to CRRT initiation was 9.9 (3.2–24.8) h.

At the start of CRRT 26 patients had an arterial blood pH<7.2, 14 a potassium >5mmol/l, 58 PaO2/FiO2 –ratio <26.6kPa indicative of at least moderate respiratory distress and 72 an hourly diuresis <0.15ml/kg/h.

### Determinants of mortality

Patients were followed up for a mean 836 days (range 1–3837 days). 81 patients (64%) died during follow-up. ICU mortality was 30%, 28-day mortality 37%, 90-day mortality 45% and one-year mortality 50%. When comparing subjects who deceased within 90 days following ICU admission to those that survived, the non-survivors had higher SOFA, SAPS-II and APACHE-II scores, were more often mechanically ventilated, had a higher norepinephrine and total vasopressor requirement and had lower hemoglobin, C-reactive protein and pH and a higher lactate level at ICU admission (Table 1). The non-surviving group also had a lower pH, bicarbonate, hourly diuresis and mean arterial pressure and higher lactate and norepinephrine requirement at CRRT initiation compared to those who were alive at 90 days (Table 3).

**Table 1.** Patient characteristics and intensive care clinical and laboratory data in the whole study population and comparisons between subgroups according to 90-day mortality.

| Variable | Whole study cohort (n = 126) | Survivors (n = 69) | Non-survivors (n = 57) | p-value |
|---|---|---|---|---|
| **Demographics and baseline data** | | | | |
| Female gender [n (%)] | 38 (30) | 21 (30) | 16 (28) | 0.77 |
| Age (years) | 64 (51–73) | 63.1 (50.7–71.7) | 67.1 (55.6–75.6) | 0.10 |
| Nosocomial infection [n (%)] | 12 (10) | 5 (7) | 7 (12) | 0.34 |
| Immunosuppression [n (%)] | 23 (18) | 6 (9) | 17 (30) | 0.002 |
| Diabetes [n (%)] | 43 (34) | 27 (39) | 16 (28) | 0.19 |
| Hypertension [n (%)] | 74 (59) | 41 (59) | 33 (58) | 0.86 |
| Pulmonary disease [n (%)] | 18 (14) | 10 (14) | 8 (14) | 0.94 |
| Coronary artery disease [n (%)] | 21 (17) | 7 (10) | 14 (25) | 0.03 |
| Peripheral arterial disease [n (%)] | 15 (12) | 6 (9) | 9 (16) | 0.22 |
| Liver cirrhosis [n (%)] | 5 (4) | 2 (3) | 3 (5) | 0.50 |
| Hematological disease [n (%)] | 9 (7) | 2 (3) | 7 (12) | 0.04 |
| Solid malignancy [n (%)] | 12 (10) | 4 (6) | 8 (14) | 0.12 |
| Baseline creatinine, n = 96 (µmol/l) | 86 (64–122) | 78 (63–138) | 96 (72–129) | 0.64 |
| Baseline eGFR, n = 96 (ml/min/1.73m2) | 84±29 | 85±23 | 81±35 | 0.49 |
| **Intensive care clinical data** | | | | |
| Peak SOFA | 14.6±3.3 | 13.7±2.9 | 15.6±3.5 | 0.0007 |
| SAPS-II | 58.8±16.3 | 53.6±14.6 | 65.3±16.2 | <0.0001 |
| APACHE-II | 26.5±7.2 | 24.4±5.9 | 29.1±7.7 | 0.0003 |
| ICU stay (days, ICU survivors, n = 88) | 9.9 (5.0–19.7) | 10.0 (5.9–21.9) | 9.7 (4.5–15.0) | 0.30 |
| Medical patients [n (%)] | 97 (77) | 55 (80) | 42 (74) | 0.42 |
| Mechanical ventilation [n (%)] | 96 (76) | 48 (70) | 48 (84) | 0.05 |
| Days on mechanical ventilation (days) | 6.9 (2.7–13.7) | 8.6 (4.6–19.8) | 4.2 (1.2–10.0) | 0.001 |
| Mean arterial pressure (mmHg) | 70 (59–80) | 71 (60–80) | 67 (58–80) | 0.65 |
| Vasopressor use [n (%)] | 120 (95) | 64 (93) | 56 (98) | 0.15 |
| Norepinephrine dose (µg/kg/min) | 0.07 (0.02–0.20) | 0.05 (0.02–0.15) | 0.11 (0.05–0.21) | 0.03 |
| Maximum noradrenalin dose (µg/kg/min) | 0.24 (0.16–0.44) | 0.21 (0.13–0.40) | 0.26 (0.18–0.51) | 0.08 |
| Number of vasopressors (n) | 1 (1–2) | 1 (1–2) | 2 (1–2) | 0.04 |
| **Laboratory data at intensive care admission** | | | | |
| Hemoglobin (g/l) | 106±18 | 110±17 | 101±19 | 0.007 |
| Leukocytes (g/l) | 12.3 (5.9–19.7) | 13.4 (7.8–21.1) | 9.8 (4.5–18.0) | 0.10 |
| Trombocytes ($E^9$/l) | 102 (53–173) | 108 (69–172) | 79 (50–173) | 0.11 |
| C-reactive protein (mg/l) | 190 (94–290) | 222 (135–333) | 150 (58–228) | 0.004 |
| Creatinine (µmol/l) n = 114 | 237 (176–339) | 254 (178–353) | 228 (176–325) | 0.28 |
| Urea (mmol/l) n = 97 | 16.8 (12.4–23.7) | 16.7 (12.9–23.7) | 17.3 (10.9–24.7) | 0.83 |
| Troponin T (ng/l) n = 95 | 79 (33–318) | 62 (34–190) | 96 (33–511) | 0.28 |
| International normalized ratio n = 113 | 1.3 (1.1–2.0) | 1.2 (1.1–1.5) | 1.6 (1.3–2.5) | 0.0003 |
| Alanine aminotransferase (IU/l) n = 103 | 62 (29–196) | 60 (28–141) | 76 (29–362) | 0.37 |
| Bilirubin (µmol/l) n = 105 | 21 (13–42) | 19 (11–38) | 25 (13–53) | 0.30 |
| pH | 7.27 (7.20–7.34) | 7.29 (7.24–7.37) | 7.23 (7.15–7.32) | 0.005 |
| Base excess | -9.3±6.2 | -8.6±5.9 | -10.2±6.4 | 0.16 |
| Bicarbonate (mmol/l) | 17.2±4.4 | 17.8±4.4 | 16.6±4.5 | 0.15 |
| Lactate (mmol/l) | 2.8 (1.6–5.9) | 2.3 (1.3–5.2) | 4.1 (2.0–8.2) | 0.004 |
| Sodium (mmol/l) | 135 (132–138) | 134 (131–138) | 136 (132–138) | 0.38 |
| Potassium (mmol/l) | 4.2 (3.8–4.8) | 4.2 (3.9–4.8) | 4.2 (3.8–4.8) | 0.55 |
| Chloride (mmol/l) | 106 (102–109) | 105 (102–108) | 107 (102–110) | 0.34 |

(*Continued*)

**Table 1.** (Continued)

| Variable | Whole study cohort (n = 126) | Survivors (n = 69) | Non-survivors (n = 57) | p-value |
|---|---|---|---|---|
| Ionized calcium (mmol/l) | 1.04±0.11 | 1.05±0.12 | 1.03±0.10 | 0.40 |

SOFA = Sequential organ failure assessment; SAPS-II = simplified acute physiology II score; APACHE-II = acute physiology and chronic health Evaluation II score.

**Table 2. Blood culture findings.**

| Pathogen | Number of patients with positive cultures (%) |
|---|---|
| *Staphylococcus aureus* | 31 (25) |
| *Escherichia coli* | 24 (19) |
| *Streptococcus pneumoniae* | 15 (12) |
| *Streptococcus pyogenes* (A) | 14 (11) |
| *Candida albicans* | 8 (6) |
| *Enterococcus faecalis* | 5 (4) |
| *Pseudomonas aeruginosa* | 5 (4) |
| *Staphylococcus epidermidis* | 4 (3) |
| *Streptococcus mitis* | 4 (3) |
| *Klebsiella pneumoniaea* | 4 (3) |
| *Streptococcus betahemolyticus* (not A) | 3 (2) |
| *Enterococcus faecium* | 3 (2) |
| *Klebsiella oxytoca* | 3 (2) |
| *Bacteroides fragilis* | 3 (2) |
| *Streptococcus dysgalactiae* | 2 (2) |
| *Neisseria meningitidis* | 2 (2) |
| *Proteus vulgaris* | 2 (2) |
| *Serratia marcescens* | 2 (2) |
| *Clostridium septicum* | 2 (2) |
| *Staphylococcus haemolyticus* | 2 (2) |
| *Candida glabrata* | 2 (2) |
| *Strepticoccus anginosus* | 1 (1) |
| *Streptococcus salivarius* | 1 (1) |
| *Enterococcus gallinarum* | 1 (1) |
| *Aerococcus viridans* | 1 (1) |
| *Haemophilus influenzae* | 1 (1) |
| *Moraxella nonliquefaciens* | 1 (1) |
| *Corynebacterium minutissimum* | 1 (1) |
| *Pseudomonas mendocina* | 1 (1) |
| *Citrobacter brakii* | 1 (1) |
| *Citrobacter freundii* | 1 (1) |
| *Raoultella planticola* | 1 (1) |
| *Proteus mirabilis* | 1 (1) |
| *Fusobacterium nucleatum* | 1 (1) |
| *Bacteroides ovatus* | 1 (1) |
| *Clostridium bifermentas* | 1 (1) |
| *Clostridium clostriforme* | 1 (1) |
| *Candida dubliensis* | 1 (1) |

**Table 3. Variables at initiation of CRRT according to 90-day mortality.**

| Variable | Survivors (n = 69) | Nonsurvivors (n = 57) | p-value |
|---|---|---|---|
| CRRT initiation after ICU admission (h) | 11.6 (3.9–27.9) | 8.7 (3.1–23.3) | 0.42 |
| Dialysis dose (ml/kg/h) | 33.7 (27.8–36.0) | 34.4 (29.9–36.1) | 0.58 |
| Creatinine (μmol/l) n = 117 | 325 (247–460) | 300 (200–391) | 0.20 |
| Urea (mmol/l) n = 100 | 20.0 (14.5–26.1) | 21.2 (15.7–33.9) | 0.38 |
| Potassium (mmol/l) | 4.1 (3.8–4.6) | 4.2 (3.9–4.7) | 0.87 |
| pH | 7.29 (7.23–7.37) | 7.27 (7.15–7.33) | 0.009 |
| Bicarbonate (mmol/l) | 18.4±3.8 | 16.3±4.2 | 0.004 |
| Lactate (mmol/l) | 1.7 (1.1–3.6) | 5.0 (2.0–7.6) | <0.0001 |
| Diuresis (ml/kg/h) | 0.15 (0.06–0.29) | 0.05 (0.02–0.24) | 0.006 |
| Fluid Balance (ml) | 2855 (470–5748) | 2395 (326–5689) | 0.52 |
| Mean arterial pressure (mmHg) | 70 (65–80) | 67 (60–75) | 0.03 |
| Norepinephrine dose (μg/kg/min) | 0.10 (0.05–0.23) | 0.17 (0.11–0.29) | 0.02 |
| PaO2/FiO2-ratio (kPa) | 28.0 (19.3–39.4) | 24.4 (16.0–38.1) | 0.19 |

CRRT = Continuous renal replacement therapy; ICU = intensive care unit.

Risk factors for mortality were assessed using univariable and multivariable Cox proportional hazards models. Age, history of heart failure, immunosuppression, peak SOFA, APACHE II and SAPS II scores, CRP, international normalized ratio, lactate, pH and bicarbonate at ICU admission and lactate, pH, bicarbonate and base excess at CRRT initiation were associated with mortality in the univariable models (S2 Table).

In the multivariable Cox proportional hazards model the significant explanatory variables at ICU admission for death were: SAPS II score (HR 1.85, 95%CI 1.35–2.52, p = 0.0001), history of heart failure (HR 2.21, 95%CI 1.29–3.81, p = 0.004), immunosuppression (HR 2.85, 95%CI 1.46–5.57, p = 0.002), pH (HR 0.78, 95%CI 0.61–0.99, p = 0.04) and C-reactive protein at ICU admission (HR 0.70, 95%CI 0.54–0.91, p = 0.007) (Fig 1). In the multivariable model for CRRT initiation the significant explanatory variables for death were: APACHE II score (HR 1.71, 95%CI 1.31–2.23, p<0.0001), history of heart failure (HR 2.78, 95%CI 1.65–4.68, p = 0.0001), immunosuppression (HR 2.42 95%CI 1.40–4.20, p = 0.002) and lactate at CRRT initiation (HR 1.57, 95%CI 1.33–1.86, p<0.0001) (Fig 1).

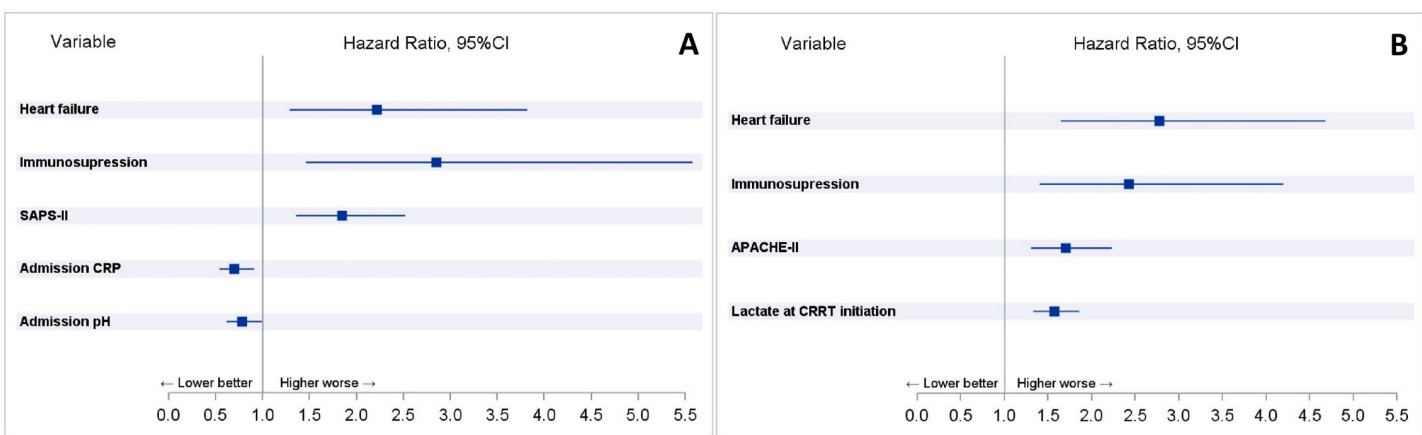

**Fig 1.** Variables independently associated with mortality in respective multivariable Cox proportional hazards models for ICU admission (Panel A) and CRRT initiation (Panel B).

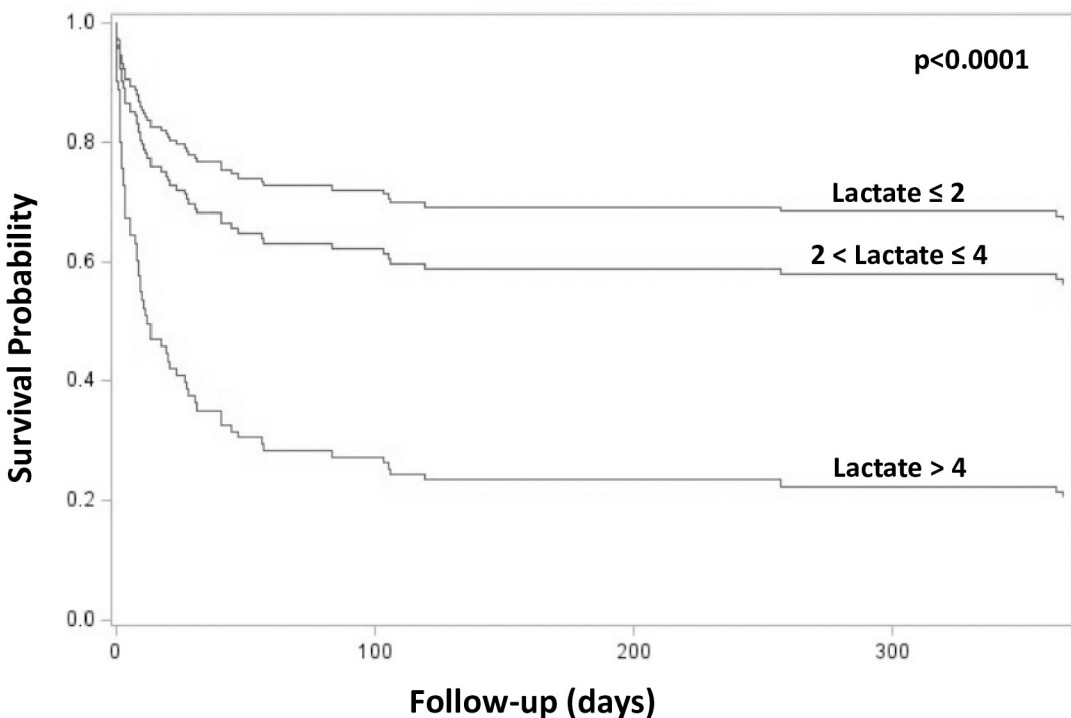

**Fig 2. Survival probability according to lactate at CRRT initiation adjusted for acute physiology and chronic health evaluation II score.**

When only patients with septic shock (n = 90) were included in the multivariable model the results remained similar. Variables independently associated with mortality in patients with septic shock were: APACHE II score (HR 1.759, 95%CI 1.303–2.376, p = 0.0002), history of heart failure (HR 2.211, 95%CI 1.178–4.152, p = 0.01), immunosuppression (HR 2.173, 95%CI 1.177–4.011, p = 0.01) and lactate at CRRT initiation (HR 1.449, 95%CI 1.206–1.742, p<0.0001).

ICU mortality was 65% and 90-day mortality 74% in patients with lactate exceeding 4 mmol/l at CRRT initiation compared to 9% and 26% in patients with lactate ≤2 mmol/l (p<0.0001, for both comparisons) (Fig 2). A lactate exceeding 4 mmol/l at CRRT initiation was chosen as a lower limit for the high lactate subgroup based on the ROC curve analysis data as a lactate of 4.3 mmol/l had a similar Youden's J index as a lactate of 5mmol/l and the best sensitivity and specificity combination (sensitivity 0.74 and corresponding specificity 0.84) for discriminating patients deceased in the ICU from ICU survivors. Lactate showed good univariate predictive power for ICU mortality in the ROC curve analyses with AUCs of 0.76 (95%CI 0.66–0.85) for lactate measured at ICU admission and 0.84 (95%CI 0.72–0.95) at CRRT initiation (Fig 3). However, for 90-day mortality only lactate at CRRT initiation showed fair predictive value with an AUC of 0.75 (95%CI 0.66–0.84), whereas, the AUC for lactate at ICU admission was rather poor 0.66 (95%CI 0.56–0.75).

Lactate clearance during the first 24 hours of CRRT (difference between lactate at CRRT initiation and 24 hours later) was not significantly associated with mortality (HR 0.96, 95%CI 0.89–1.05, p = 0.36) although lactate measured at every time point during CRRT (6h, 12h, 18h, 24h and 48h) were positively associated with mortality (HRs ranging between 1.12–1.67 and p<0.0001 for all respective univariate Cox models). Fig 4 shows the development of lactate during CRRT according to 90-day mortality. Lactate values were significantly higher at every

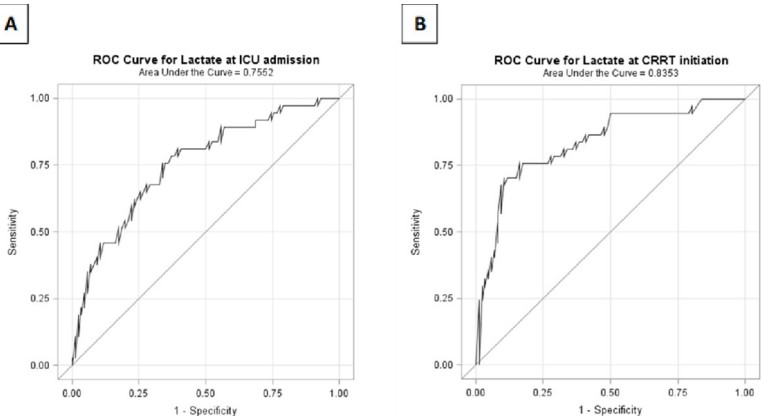

**Fig 3.** Area under the curve (AUC) of receiver operating characteristics curve (ROC) analyses for lactate at ICU admission (panel A) and lactate at CRRT initiation (panel B) in relation to ICU mortality.

time point during the first 48h of CRRT in the patients who deceased by day 90 compared to others.

Only five (8%) of the surviving patients remained dialysis-dependent at 6 months. Estimated GFR remained attenuated in survivors at 90 days (n = 52, baseline: 86±21 vs. 90-days 78±28 ml/min/1.72m$^2$, p = 0.01) and at one-year (n = 30, baseline: 84±19 vs. 90-days 70±24 ml/min/1.72m$^2$, p = 0.0007) compared to baseline values.

## Discussion

The present study shows for the first time that blood lactate at CRRT initiation is independently associated with overall mortality in patients with blood culture positive sepsis and coincident severe AKI requiring CRRT. Furthermore, lactate measured at ICU admission and at CRRT initiation were both higher in patients who died within 90 days of ICU admission and

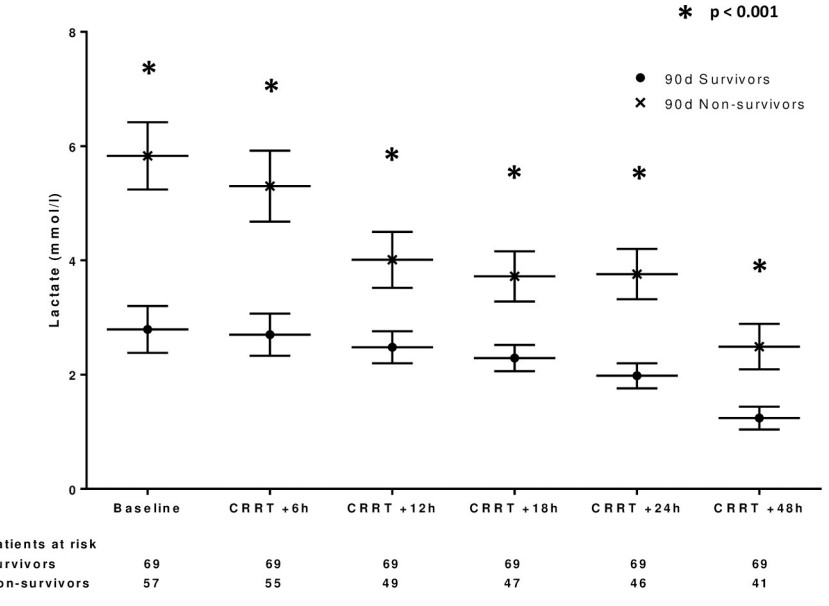

**Fig 4. Lactate clearance during CRRT according to 90-day mortality.**

lactate at CRRT initiation showed good predictive power for discriminating those who died in the ICU.

Several previous studies have explored risk factors associated with mortality in RRT dependent sepsis-associated AKI but most studies have defined sepsis according to the Sepsis-3 (suspected or shown infection with an increase in SOFA score ≥2) [5] or earlier consensus criteria, whereas, blood culture findings have seldom been described. One former study reported blood culture findings in ICU patients with RRT, but the number of patients was very limited, and the patient group was not restricted to sepsis-associated AKI [6]. The ICU mortality of 30% in our current study was similar to that reported in larger series of patients with culture-positive or culture-negative septic shock in Europe [7]. Some [8,9], but not all [10] former studies have observed a difference in outcomes between culture-positive and culture-negative septic shock. In a previous study, patients with a clinical diagnosis of sepsis based on the former 2001 International Sepsis Definitions Conference criteria (without blood culture data) [11] and coincident CRRT-dependent AKI had similar mortality compared to non-septic AKI patients on CRRT [12]. The rate of positive blood culture findings in patients considered septic has varied between 40–70% in previous studies [8,13–15]. To our best knowledge only a single previous South Korean study has reported blood culture data in a cohort of 210 CRRT patients with sepsis but even in that cohort microbial etiology was not documented in 29% of included patients and 53.8% had a malignancy [16].

Sepsis-associated hyperlactatemia is the result of stress-induced accelerated anaerobic metabolism, impaired tissue oxygen delivery and extraction, peripheral shunting, increased adrenergic stimulation and to a lesser degree impaired lactate clearance [17]. Hyperlactatemia is indisputably associated with increased mortality in broad patient populations with sepsis and septic shock [18], but data on patients with blood culture positive septic AKI requiring CRRT are scarce [19]. Passos and coworkers examined the association between lactate, lactate clearance and mortality in 186 patients with sepsis according to the formerly used systemic inflammatory response syndrome (SIRS) based criteria and coincident AKI requiring CRRT [20]. In their study lactate clearance during the first 24 hours of CRRT treatment and lactate at 24 hours were independently associated with 48-hour and 28-day mortality, but lactate at CRRT initiation was not. Only 24% of the study patients had a blood stream infection as opposed to the present study. Our current findings show that in blood culture positive sepsis patients both lactate at ICU admission and CRRT initiation are predictive of ICU mortality and lactate at CRRT initiation is also predictive of long-term mortality at least up to 90 days. Lactate clearance during the first 24 hours of CRRT was not associated with mortality despite lactate levels were measured at several time points during the first 48 hours of CRRT. This may partly be a result from the fact that many of the patients without lactate clearance in the beginning of CRRT died shortly after or even within 24 hours from CRRT initiation. CRRT potently extracts lactate when its production is not ongoing and massive. In the absence of lactate clearance and coincident clinical deterioration during the first 24 hours of CRRT the benefit of continuing CRRT becomes questionable. Our current finding that lactate measured at CRRT initiation is highly predictive of mortality in spite of lactate clearance during the first 24 hours of CRRT shifts risk prediction to an earlier stage. Furthermore, our current findings show that only 26% of blood culture positive sepsis patients with lactate exceeding 4 mmol/l at CRRT initiation are alive at 90 days follow-up. This emphasizes the importance of considering lactate levels for risk-stratification purposes when making decisions on initiating CRRT in critically ill sepsis patients with AKI.

We identified 38 different causative pathogens for sepsis and associated AKI in the present study. The bacteria with the highest incidence were *Staphylococcus aureus*, *Escherichia coli*, *Streptococcus pneumoniae* and *Streptococcus pyogenes*. Only one patient had septicemia caused

by an extended-spectrum beta-lactamase producing strain of *Escherichia coli*. Methicillin or carbapenem resistant strains of bacteria were not observed. The blood culture findings were similar to those previously reported in septic emergency department patients in Finland [21]. C-reactive protein at ICU admission was surprisingly independently and inversely associated with mortality in this selected and comorbid patient population which is in contrast with previous studies in broad sepsis patients [22]. Immunosuppression, history of heart failure and APACHE II and SAPS II scores were also independently associated with mortality in the Cox models.

At 6 months ca. 8% of the surviving patients were on maintenance dialysis and eGFR remained attenuated in survivors at 90-days and at one-year following sepsis-associated AKI and CRRT. These findings are in line with previous reports showing a high incidence of chronic kidney disease in primary survivors of septic AKI [23].

The limitations of this study pertain to its retrospective design and limited sample size. However, the study population was limited to patients with true blood culture positive sepsis, AKI and CRRT treatment compared to former studies in septic AKI and RRT with more heterogenous patient populations due to the definition of sepsis. Since data of the current study were collected at a single tertiary medical center in a developed country, the results concerning blood culture data may not apply to other institutions in other geographical areas. The prevalence of extended spectrum antibiotic resistance is extremely low in Finland, which, was also observed in the blood culture positive pathogens of this study. Nevertheless, the findings were quite distinct, and a limited sample size is not likely to detract from the validity of the main findings of this study concerning the associations between lactate and early and late mortality and its predictive value for ICU mortality.

## Conclusions

Our study shows for the first time that lactate measured at CRRT initiation is predictive of ICU mortality and independently associated with overall mortality in patients with blood culture positive sepsis and AKI requiring CRRT. Microbial etiology for septic AKI requiring CRRT is diverse.

## Supporting information

**S1 Fig. Flow-chart of the study.**
(PPTX)

**S1 Table. Antimicrobial regimens used: 1st empiric treatment; specific treatment; and all antimicrobial regimes used during intensive care unit (ICU) stay.**
(DOCX)

**S2 Table. Factors associated with mortality in univariate models.** For comparability, the hazard ratios are for standardized variables with mean 0 and SD 1.
(DOCX)

## Acknowledgments

The authors are grateful to Mrs Eveliina Loikas, RN, for help with the data collection and to Mrs Noora Kartiosuo, MSc, for statistical consultation.

## Author Contributions

**Conceptualization:** Mikko J. Järvisalo, Tapio Hellman, Panu Uusalo.

**Data curation:** Mikko J. Järvisalo, Tapio Hellman, Panu Uusalo.

**Formal analysis:** Mikko J. Järvisalo.

**Investigation:** Mikko J. Järvisalo, Panu Uusalo.

**Methodology:** Mikko J. Järvisalo, Tapio Hellman.

**Project administration:** Mikko J. Järvisalo.

**Visualization:** Tapio Hellman, Panu Uusalo.

**Writing – original draft:** Mikko J. Järvisalo.

**Writing – review & editing:** Tapio Hellman, Panu Uusalo.

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
