## [Decision Letter · Decision Letter 0]

15 Feb 2021

PONE-D-21-00589

Mortality and Associated Risk Factors in Patients with Blood Culture Positive Sepsis and Acute Kidney Injury Requiring Continuous Renal Replacement Therapy

PLOS ONE

Dear Dr. Järvisalo,

Thank you for submitting your manuscript to PLOS ONE. After careful consideration, we feel that it has merit but does not fully meet PLOS ONE’s publication criteria as it currently stands. Therefore, we invite you to submit a revised version of the manuscript that addresses the points raised during the review process.

We look forward to receiving your revised manuscript.

Kind regards,

Aleksandar R. Zivkovic

Academic Editor

PLOS ONE

Note: HTML markup is below. Please do not edit.]

Reviewers' comments:

Reviewer's Responses to Questions

**Comments to the Author**

1. Is the manuscript technically sound, and do the data support the conclusions?

Reviewer #1: Yes

Reviewer #2: Yes

Reviewer #3: Partly

Reviewer #4: Partly

Reviewer #5: No

2. Has the statistical analysis been performed appropriately and rigorously? 

Reviewer #1: Yes

Reviewer #2: Yes

Reviewer #3: I Don't Know

Reviewer #4: Yes

Reviewer #5: No

3. Have the authors made all data underlying the findings in their manuscript fully available?

Reviewer #1: Yes

Reviewer #2: Yes

Reviewer #3: No

Reviewer #4: Yes

Reviewer #5: Yes

4. Is the manuscript presented in an intelligible fashion and written in standard English?

Reviewer #1: Yes

Reviewer #2: Yes

Reviewer #3: Yes

Reviewer #4: Yes

Reviewer #5: Yes

5. Review Comments to the Author

Reviewer #1: Thank you for the opportunity to review this interesting manuscript. In general, it is very well written and I can only propose minor changes:

Retrospective nature is first mentioned just before “Methods” in the main text (missing in title and abstract)

Flowchart is missing with information on e.g. how many ICU patients were screened, how many were excluded and how many could be evaluated.

Further it is unclear to me, at which point could the blood cultures be positive and/or CRRT administered (admission or later on, simultaneously or could CRRT be begun days after ascertained bacteremia). What is the diagnostic protocol for blood cultures in your hospital 2x2+2 for fungi or 3x2 or orther approach? Also, how many blood culture needed to be positive for diagnosis of an infection?

Data on time from hospital/ICU admission to positive blood culture diagnosis is missing.

I would prefer exact numbers on SOFA, SAPS-II and APACHE instead of their description in the “Results”-Section (i.e. “All of the patients were critically ill, with high SOFA, Simplified acute physiology II (SAPS-II) and Acute physiology and chronic health evaluation II (APACHE-II) scores”)

Could you additionally provide 28-day mortality to enable further comparison for future studies?

I find your limitations section very harsh on your part. Over one-hundred patients with bacteremia and septic-shock with AKI is a respectable size. Of course you don’t have to modify this part; however I think, that apart from the retrospective nature, that makes causality investigation more difficult, there is no further limitation.

I would also recommend to place the outlook of the importance of you findings a bit more prominent (“how do your results impact clinical practice”).

I would further recommend to improve the readability of Table 1, since it is not yet intuitive in its interpretation.

Lastly, I would like you to consider, whether, due to the fact that 98% of your patients are septic shock patients with AKI, you would like to focus just on those patients and change the title accordingly. This way your manuscript would become even more specific and I do not see a benefit from keeping the 3 non-shock patients in the analysis.

Reviewer #2: This manuscript does a good job in a retrospective population to recognize risk factors associated with mortality in AKI patients undergoing CRRT with blood culture positive sepsis. In the retrospective, register-based study, a cohort of 126 patients were followed up for a mean 836 days. Comprehensive laboratory and clinical data were gathered and studied. The result shows that lactate measured at ICU admission and CRRT initiation are predictive of ICU mortality and lactate at CRRT initiation is independently associated

with overall mortality in patients. It has some predictive value for early risk stratification in this specific populations.

With regard to the data，I will make some concrete comments below.

1.As the author notes, the sample size of the study was relatively small, and the patients were all carried out in a single center. The results may be influenced by differences in bacterial profiles in different regions.

2.Lactate clearance is a significant indicator of sepsis prognosis, and it would be convincing if the data could be analyzed.

3.As a more minor point with regard to the inverse correlation between CRP and mortality. This is somewhat confusion. If you can have any data on other inflammatory factors? Procalcitonin is more specific than C-reactive protein in sepsis, and it may be more meaningful to analysis these data.

I hope my comments make sense. Please do not hesitate to contact me if not.

Reviewer #3: The manuscript «Mortality and Associated Risk Factors in Patients with Blood Culture Positive Sepsis and Acute Kidney Injury Requiring Continuous Renal Replacement Therapy” by Järvisalo and co-workers is submitted to PLOS one for consideration for publication. This retrospective cohort analysis in patients with blood-culture positive septic shock requiring early renal replacement (CRRT) therapy reveals that lactate measured at ICU admission as well as at CRRT initiation are predictive for ICU mortality.

The authors claim that this is a unique finding. As this may well be so in this specific subgroup of patients with septic shock, multiple studies have shown that elevated lactate in patients with septic shock in general is associated with increased mortality (e.g. Casserly et al, CCM 2015). Moreover, the reported ICU mortality of 30% in this study reflects average mortality of septic shock in ICUs in Europe (Vincent et al, SOAP study, CCM 2006) including culture-positive and culture-negative sepsis. Furthermore, studies have not shown any difference in outcome between culture-positive and culture-negative septic shock (most recent: Kim et al, Crit Care 2021). Thus, this is a subgroup-analysis of a specific group of patients with septic shock (culture-positive and AKI requiring CRRT) and therefore, these results seem not surprising, and in my opinion not game-changing.

If considered for publication, I have some suggestions to improve and strengthen the manuscript.

1. Both, Table 1 and 3 have similar results, they might be merged into one table (whole population/survivors/non-survivors).

2. Table 2 and Supplemental Table 1: It would help the reader, if percentage is given (according to Table 1). Please specify in Supplemental Table 1, at what stage these antimicrobial regimens were used (at admission, throughout the ICU stay, throughout hospital stay?). Some of them seem very narrow to be the first-line antimicrobial agent for the initial treatment of septic shock.

3. One of the main conclusions of the study, that lactate is predictive for ICU mortality, is only shown in the univariate analysis. However, this does not proof a causal relationship and therefore this conclusion should be softened. This statement is also the opening of discussion, however the results only indicate that lactate at the time of CRRT initiation is independendly associated with mortality. This sentence should therefore be clarified.

4. “In a way our current findings shift risk prediction to an earlier stage, which can be considered valuable for clinical purposes”: However, at this early stage (at ICU admission), the blood culture results are often unknown. As in a majority of patients in the present study, CRRT has been started within 12 hours after admission; the results of the blood cultures are most probably not available at CRRT initiation either. This is a limitation of the use of this “early” predictor of mortality as it seems only useful in retrospect.

5. It would be helpful for the reader to add (or replace) one figure to show the main results of the study, namely the multivariate analysis at both times (ICU admission and CRRT initiation).

6. In Figure 1, the numbers and legends are very hard to read due to the size. Both, A and B show mortality by lactate level. However, these results are not adjusted and show, that the sicker they are, the higher the lactate and consecutively the higher the mortality.

Reviewer #4: The study investigated, ID number PONE-D-21-00589 and “Mortality and Associated Risk Factors in Patients with Blood Culture Positive Sepsis and Acute Kidney Injury Requiring Continuous Renal Replacement Therapy” titled I think it is an exciting study in terms of its results. This research is good, but acceptable if the following recommendations are taken into account.

1. Are there criteria for not accepting patients in the study?

2. Do bacteria grown in blood culture have criteria for acceptance as causative agents?

3. Are there criteria for accepting the bacteria growing in the blood culture as the cause of sepsis? and Has the source been investigated for these bacteria that cause sepsis?

4. Is there a correlation between the bacteria causing sepsis and mortality?

5. What were the drugs that were started empirically in the treatment of sepsis as monotherapy or in combination?

6. How many patients' treatments were changed to specific treatment after the agent was produced in blood culture?

7. How many days empirical and specific drug treatments were given.

8. It is recommended to write the names of bacteria in accordance with medical spelling rules.

Reviewer #5: Järvisalo et al present a study of 126 patients with cultures positive sepsis and AKI and examined predictors of 90 day mortality. They found that elevated lactate level at admission and initiation of CRRT were associated with higher mortality level. The study has several shortcomings.

Major points:

1- The main problem I have with the study and the conclusion is I don't think the authors can truly say that this is the first paper that showed that elevated lactate was associated with higher mortality. I think this has been shown in numerous studies before.

2- I have several reservation about the statistical analysis. The authors selected patients with sepsis based on the Sep3 definition which takes into account SOFA score. Than the authors adds two more severity of illness score ( APACHE and SAPS) in the analysis. First, the is the problem of over-fitting when using so many severity of illness score. Second, since SOFA score was part of the SEP3 definition and therefore inclusion in the study, I am not sure it should be included in the multivariate. analyses.

3- I am having a hard time understanding the rationale for choosing lactate of 2 and 4 in the analysis. The authors should explain why those were chosen: was it based on the ROC? was is based on the univariate analyses? was is based on prior studies?

I am not sure this study adds much to the literature beyond what we already know about lactate levels in critically ill patients.

6. PLOS authors have the option to publish the peer review history of their article (what does this mean?). If published, this will include your full peer review and any attached files.

Reviewer #1: **Yes: **Gregor A. Schittek, Dr. med.

Reviewer #2: No

Reviewer #3: No

Reviewer #4: No

Reviewer #5: No

---

## [Author Response · Author response to Decision Letter 0]

17 Mar 2021

Academic Editor 

Aleksandar R. Zivkovic

PLOS ONE

RE: MS# PONE-D-21-00589 “Mortality and Associated Risk Factors in Patients with Blood Culture Positive Sepsis and Acute Kidney Injury Requiring Continuous Renal Replacement Therapy”

Dear Professor Zivkovic,

Thank you for your letter of February 15, 2021 concerning our manuscript. In our study we examined risk factors for mortality in a cohort of patients with blood culture positive sepsis and acute kidney injury (AKI) requiring continuous renal replacement therapy (CRRT). We believe that we have been able to respond to all the points raised by the reviewers as outlined below and feel that the manuscript has improved with the changes made. We are therefore grateful for the opportunity to submit the revised manuscript and hope that you will find it acceptable for publication in PLOS ONE.

Technical issues:

1. We have made corrections in the style of the revised manuscript to meet PLOS ONE's style requirements, including those for file naming. 

2. On the topic of data availability: The authors support data availability. However, as for this retrospective register-based study the institutional review board waived the need for informed consent in terms of data collection and analysis and publication of results but together with the national Office of the Data Protection Ombudsman did not allow for public distribution of data as it would be in violation of Finnish laws (Tietosuojalaki 5.12.2018/1050 31§) without informed consent even when personal identification codes (social security codes) are removed. Furthermore, the data contain potentially sensitive information. Concerning ethics: as the number of patients included in the study is relatively low and the data would need to include dates of ICU admission, gender and age of the patients to reproduce some of the analyses performed in the study, we feel that the risk of identifying individual patients from the data set is increased. 

Acquiring informed consent for the study was not applicable since the data were gathered for 2010-2019 and therefore majority of the patients are currently deceased. Data that support the findings of the current study are, however, available from the data sets of the Department of Anesthesiology and Intensive Care and the Informatics Department of Turku University Hospital on reasonable request after permission of the Ethics Committee of Southwest Finland Hospital District. The data request may be made to the Ethics Committee of Southwest Finland Hospital District, eettinen.toimikunta@tyks.fi. 

Summary of data restrictions:

-The data restrictions have been imposed by the institutional research ethics review board and the National Office of the data protection ombudsman.

-According to Finnish laws the data cannot be publicly distributed. Furthermore, the data contain potentially sensitive information and the risk of identifying individual patients from the data set is increased.

-The data request may be made to the Ethics committee of Southwest Finland hospital district, eettinen.toimikunta@tyks.fi.

The data availability statement has been revised in cooperation with the editorial office: “The data underlying this study contain potentially identifying participant information and cannot be shared publicly. Future data access requests should be sent to the Ethics Committee of Southwest Finland Hospital District (eettinen.toimikunta@tyks.fi) or the Department of Anesthesiology and Intensive Care and the Informatics Department of Turku University Hospital via the corresponding author.”

3. The ethics statement has been moved to the Methods section in the revised manuscript as asked. 

All authors have read the “Instructions to authors” and approved submission of the submitted version of the manuscript. The material is original and the manuscript has not been published and is not being considered for publication elsewhere. The authors do not have any conflicts of interest in connection with the submitted article. If accepted, the paper will not be published elsewhere in the same form, in English or in any other language, without written consent of the copyright holder.

MJJ, TH and PU designed the study and were responsible for the data collection. MJJ performed the

statistical analyses. MJJ drafted the manuscript. TH and PU revised the manuscript.

Yours sincerely,

Adjunct professor Mikko Järvisalo, MD, PhD

Consultant in nephrology, internal medicine and intensive care

Intensive care unit, Turku University Hospital

Building 18, TG3B

PO Box 52, 20521 Turku, Finland

Email: mikko.jarvisalo@tyks.fi

Phone: +358-2-3130049 

 

Reviewers' comments:

Reviewer #1: Thank you for the opportunity to review this interesting manuscript. In general, it is very well written and I can only propose minor changes:

Retrospective nature is first mentioned just before “Methods” in the main text (missing in title and abstract)

We have revised the title of the manuscript to include “A Retrospective Study” and included this information to the Abstract of the revised manuscript.

Flowchart is missing with information on e.g. how many ICU patients were screened, how many were excluded and how many could be evaluated.

We have included a flow chart of the study (Supplemental Figure 1) as well as a more detailed description on patients included in the study and patients excluded (Methods: page 4, paragraph 1, lines 5-12).

Further it is unclear to me, at which point could the blood cultures be positive and/or CRRT administered (admission or later on, simultaneously or could CRRT be begun days after ascertained bacteremia). What is the diagnostic protocol for blood cultures in your hospital 2x2+2 for fungi or 3x2 or orther approach? Also, how many blood culture needed to be positive for diagnosis of an infection?

Blood cultures are acquired at our institution usually at hospital admission 2x1 and at ICU admission 2x1 and thereafter at temperature spikes of ≥38°C (when previous blood cultures remain unanswered or negative). 

We have included data on the timing of diagnosis (positive blood culture(s)) in relation to ICU admission and CRRT initiation in the revised manuscript (Results: page 7, paragraph 1, lines 3-5). The median time from the positive blood culture(s) to the CRRT initiation was 0.5 (-0.6 – 1.7) days. 

As stated in the manuscript, all of the blood culture findings had been examined case by case and considered to be relevant etiological agents for the sepsis present by the ICU infectious diseases senior consultant (a certain number of positive cultures was not required) (Methods: page 4, paragraph 1, lines 7-11). We have also included data on the number of positive blood cultures for every patient in the revised manuscript (Results: page 7, paragraph 1, lines 5-6). The median number of blood cultures per patient was 2 (2-4) which is probably a comparatively low rate in this cohort due the fact that cultures were found positive for each patient. The median number of positive blood cultures was 2 (2-2). 

Data on time from hospital/ICU admission to positive blood culture diagnosis is missing.

The median time from ICU admission to diagnosis (positive blood culture(s)) was 0.3 (-0.6 – 1.3) days. We have included these data to the revised manuscript as suggested (Results: page 7, paragraph 1, lines 3-5)

I would prefer exact numbers on SOFA, SAPS-II and APACHE instead of their description in the “Results”-Section (i.e. “All of the patients were critically ill, with high SOFA, Simplified acute physiology II (SAPS-II) and Acute physiology and chronic health evaluation II (APACHE-II) scores”)

We have revised this sentence in the revised manuscript (Results: page 7, paragraph 2, lines 13-14)

Could you additionally provide 28-day mortality to enable further comparison for future studies?

The 28-day mortality was 37.3%. We have included data on 28-day mortality in the revised manuscript (Results: page 7, paragraph 4, line 23).

I find your limitations section very harsh on your part. Over one-hundred patients with bacteremia and septic-shock with AKI is a respectable size. Of course you don’t have to modify this part; however I think, that apart from the retrospective nature, that makes causality investigation more difficult, there is no further limitation.

We thank the reviewer for the kind comment, however, as some reviewers point out that the sample size was low, we have not revised the limitations section in this regard. 

I would also recommend to place the outlook of the importance of you findings a bit more prominent (“how do your results impact clinical practice”).

Again, we thank the reviewer for the kind comment, however, as some reviewers point out that the conclusions are too prominent we have revised the conclusions accordingly (Discussion: page 10, paragraph 2, lines 24-27 and page 11, paragraph 1, lines 1-8).

I would further recommend to improve the readability of Table 1, since it is not yet intuitive in its interpretation.

We have merged Tables 1 and 3 (as it was suggested by another reviewer) and aimed to improve the readability of the data in the new Table 1 by including subtitles in the table (Table 1).

Lastly, I would like you to consider, whether, due to the fact that 98% of your patients are septic shock patients with AKI, you would like to focus just on those patients and change the title accordingly. This way your manuscript would become even more specific and I do not see a benefit from keeping the 3 non-shock patients in the analysis.

There was a very unfortunate error in the manuscript concerning the number/proportion of patients with septic shock. 95% of the patients required vasopressors but only 90 patients (71%) were considered to have septic shock according to the sepsis-3 criteria. We have corrected these data in the revised manuscript (Results: page 7, paragraph 2, line 16). 

As we find the sample size limited, we are not eager to limit it further by only including patients with septic shock. The results concerning the final multivariable model for CRRT initiation, however, remained similar when only patients with septic shock were included in the model. Variables independently associated with mortality in patients with septic shock were: lactate at CRRT initiation (HR 1.449, 95%CI 1.206-1.742, p<0.0001), APACHE II score (HR 1.759, 95%CI 1.303-2.376, p=0.0002), history of heart failure (HR 2.211, 95%CI 1.178-4.152, p=0.01) and immunosuppression (HR 2.173, 95%CI 1.177-4.011, p=0.01). We thank the reviewer for helping us notice and correct the error. We have included these data in the revised manuscript (Results: page 8, paragraph 4, lines 17-21).

 

Reviewer #2: This manuscript does a good job in a retrospective population to recognize risk factors associated with mortality in AKI patients undergoing CRRT with blood culture positive sepsis. In the retrospective, register-based study, a cohort of 126 patients were followed up for a mean 836 days. Comprehensive laboratory and clinical data were gathered and studied. The result shows that lactate measured at ICU admission and CRRT initiation are predictive of ICU mortality and lactate at CRRT initiation is independently associated

with overall mortality in patients. It has some predictive value for early risk stratification in this specific populations.

With regard to the data，I will make some concrete comments below.

1. As the author notes, the sample size of the study was relatively small, and the patients were all carried out in a single center. The results may be influenced by differences in bacterial profiles in different regions.

We have discussed these limitations in the Discussion section of the revised manuscript (Discussion page 11, paragraph 4, lines 22-27)

2. Lactate clearance is a significant indicator of sepsis prognosis, and it would be convincing if the data could be analyzed.

We have included data on lactate clearance during CRRT in the revised manuscript, including a figure. Lactate clearance during the first 24 hours of CRRT was not associated with mortality in a Cox proportional hazards model although lactate measurements on several time points during the first 48 hours of CRRT were. We have included these data in the revised manuscript (Results: page 9, paragraph 2, lines 7-13 and Figure 4) and also included a short discussion on the matter (Discussion: page 10, paragraph 2, lines 24-27 and page 11, paragraph 1, lines 1-8). 

3. As a more minor point with regard to the inverse correlation between CRP and mortality. This is somewhat confusion. If you can have any data on other inflammatory factors? Procalcitonin is more specific than C-reactive protein in sepsis, and it may be more meaningful to analysis these data.

Unfortunately, we have no data available for procalcitonin. 

 

Reviewer #3: The manuscript «Mortality and Associated Risk Factors in Patients with Blood Culture Positive Sepsis and Acute Kidney Injury Requiring Continuous Renal Replacement Therapy” by Järvisalo and co-workers is submitted to PLOS one for consideration for publication. This retrospective cohort analysis in patients with blood-culture positive septic shock requiring early renal replacement (CRRT) therapy reveals that lactate measured at ICU admission as well as at CRRT initiation are predictive for ICU mortality.

The authors claim that this is a unique finding. As this may well be so in this specific subgroup of patients with septic shock, multiple studies have shown that elevated lactate in patients with septic shock in general is associated with increased mortality (e.g. Casserly et al, CCM 2015). Moreover, the reported ICU mortality of 30% in this study reflects average mortality of septic shock in ICUs in Europe (Vincent et al, SOAP study, CCM 2006) including culture-positive and culture-negative sepsis. Furthermore, studies have not shown any difference in outcome between culture-positive and culture-negative septic shock (most recent: Kim et al, Crit Care 2021). Thus, this is a subgroup-analysis of a specific group of patients with septic shock (culture-positive and AKI requiring CRRT) and therefore, these results seem not surprising, and in my opinion not game-changing.

The reviewer is right that previous data are available that lactate is associated with mortality in general patients with septic shock according to Sepsis-3 criteria. However, to our knowledge this finding has not been previously observed in patients with blood culture positive septicemia and CRRT dependent AKI. To increase the novelty of our revised manuscript we have also included data on lactate clearance during the first 48 hours CRRT (Results: page 9, paragraph 2, lines 7-13; Figure 4; Discussion: page 10, paragraph 2, lines 24-27 and page 11, paragraph 1, lines 1-8). Furthermore, we have included the references suggested by the reviewer to the revised manuscript and revised the text accordingly (Discussion: page 9, paragraph 4, lines 19-23; page 10, paragraph 1, lines 2-5 and paragraph 2, lines 15-17; References: 7, 10 and 18).

1. Both, Table 1 and 3 have similar results, they might be merged into one table (whole population/survivors/non-survivors).

We have merged Tables 1 and 3 as suggested (Table 1)

2. Table 2 and Supplemental Table 1: It would help the reader, if percentage is given (according to Table 1). Please specify in Supplemental Table 1, at what stage these antimicrobial regimens were used (at admission, throughout the ICU stay, throughout hospital stay?). Some of them seem very narrow to be the first-line antimicrobial agent for the initial treatment of septic shock.

We have included percentages in Table 2 and Supplemental Table 1. We have also included more specific data on the antimicrobial regimen used (1st empiric treatment; specific treatment; and all antimicrobial regimes used during ICU stay (Supplemental Table 1).

3. One of the main conclusions of the study, that lactate is predictive for ICU mortality, is only shown in the univariate analysis. However, this does not proof a causal relationship and therefore this conclusion should be softened. This statement is also the opening of discussion, however the results only indicate that lactate at the time of CRRT initiation is independendly associated with mortality. This sentence should therefore be clarified.

We have revised the conclusions and the beginning of the discussion section of the revised manuscript accordingly (Abstract: page 2, paragraph 4, lines 15-17; Discussion: page 9, paragraph 4, lines 19-23 and page 12, paragraph 2, lines 6-8). 

4. “In a way our current findings shift risk prediction to an earlier stage, which can be considered valuable for clinical purposes”: However, at this early stage (at ICU admission), the blood culture results are often unknown. As in a majority of patients in the present study, CRRT has been started within 12 hours after admission; the results of the blood cultures are most probably not available at CRRT initiation either. This is a limitation of the use of this “early” predictor of mortality as it seems only useful in retrospect.

We have included data concerning the timing of diagnosis (positive blood culture(s)) in the revised manuscript. The median time from ICU admission to diagnosis (positive blood cultures) was 0.3 (-0.6 – 1.3) days and the median time from the positive blood culture(s) to CRRT initiation was 0.5 (-0.6 – 1.7) days (Results: page 7, paragraph 1, lines 3-5).

Furthermore we have rewritten the clinical implications part of the manuscript to better reflect our findings and their possible implications including discussion on lactate measured at CRRT initiation and clearance during CRRT (page 10, paragraph 2, lines 24-27 and page 11, paragraph 1, lines 1-8).

5. It would be helpful for the reader to add (or replace) one figure to show the main results of the study, namely the multivariate analysis at both times (ICU admission and CRRT initiation).

We have included a new figure designated as Figure 1 (multivariable model results) (Figure 1).

6. In Figure 1, the numbers and legends are very hard to read due to the size. Both, A and B show mortality by lactate level. However, these results are not adjusted and show, that the sicker they are, the higher the lactate and consecutively the higher the mortality.

We have omitted panels A and B from the former Figure 1 (now designated as Figure 2 in the revised manuscript) to improve readability (and as the reviewer duly pointed out both panels showed mortality by lactate level) and we have also replaced panel C with a new figure depicting survival probability in the lactate subgroups adjusted for APACHE-II (Figure 2). 

 

Reviewer #4: The study investigated, ID number PONE-D-21-00589 and “Mortality and Associated Risk Factors in Patients with Blood Culture Positive Sepsis and Acute Kidney Injury Requiring Continuous Renal Replacement Therapy” titled I think it is an exciting study in terms of its results. This research is good, but acceptable if the following recommendations are taken into account.

1.Are there criteria for not accepting patients in the study?

We have included a flow chart of the study (Supplemental Figure 1). Only patients with blood culture positive sepsis according to Sepsis-3 criteria were included in the study. Blood culture findings had been examined case by case and considered to be relevant etiological agents for the sepsis present by the ICU infectious diseases senior consultant. Patients without positive blood cultures with/or without sepsis were excluded as were patients with maintenance dialysis dependency prior to ICU admission (n=22). In one patient with a single blood culture positive for Staphylococcus capitis, the culture finding had been considered a contamination, not a causative agent for the sepsis by the infectious diseases specialist and a microbiologist and the patient was excluded from the study. (Methods: page 4, paragraph 1, lines 5-12).

2. Do bacteria grown in blood culture have criteria for acceptance as causative agents?

As stated in the manuscript all of the blood culture findings had been examined case by case and considered to be relevant etiological agents for the sepsis present by the ICU infectious diseases senior consultant (Methods: page 4, paragraph 1, lines 7-11).

3. Are there criteria for accepting the bacteria growing in the blood culture as the cause of sepsis? and Has the source been investigated for these bacteria that cause sepsis?

As stated in the manuscript all of the blood culture findings were examined case by case and considered to be relevant etiological agents for the sepsis present by the ICU infectious diseases senior consultant (Methods: page 4, paragraph 1, lines 7-11). The source of infection was blood-born in 56 (44%), skin/soft tissues in 35 (28%), urinary in 14 (11%), abdominal in 11 (9%) and respiratory in 10 (8%) patients, respectively. We have included data on the source of infection to the revised manuscript (Results; page 7, paragraph 1, lines 6-8).

4. Is there a correlation between the bacteria causing sepsis and mortality?

Unfortunately, the bacterial/fungal etiologies were so diverse and the sample size limited that this could not be reliably assessed. We have, however, examined the association between the most prevalent blood culture findings and mortality although it is clear that the study was not adequately powered to make any definite conclusions on this matter and therefore these data is not included in the revised manuscript. The 90-day mortality rates were Staphylococcus aureus: 52%; Escherichia coli: 36%; Streptococcus pneumoniae 53%, Streptococcus pyogenes: 21%, any fungemia: 56% and other bacteria 49% (p>0.15 for all comparisons). Furthermore, no differences in mortality were observed between patients with a single identified microbe (mortality 44%) or more than one microbes (mortality 57%) (p=0.34) positive in the blood cultures, but again, the study was not adequately powered to make these conclusions.

5. What were the drugs that were started empirically in the treatment of sepsis as monotherapy or in combination?

We have included more specific data on the antimicrobial regimens in the revised Supplemental Table 1 including data on empirically started antimicrobial agents (Supplemental Table 1)

6. How many patients' treatments were changed to specific treatment after the agent was produced in blood culture?

The treatment remained unchanged in 49 (39%) patients, in 28 (22%) patients an additional antimicrobial regimen was added to the treatment and in 49 (39%) patients the treatment was changed entirely. We have included these data to the revised manuscript (Results; page 7, paragraph 1, lines 9-11).

7. How many days empirical and specific drug treatments were given.

The median time from the start of first empirical therapy to the initiation of specific antimicrobial therapy based on blood culture findings was 1 (1-2.5) days. We have included these data to the revised manuscript (Results: page 7, paragraph 1, lines 11-12). Unfortunately, we do not have data on the total duration of specific antimicrobial therapy as surviving patients continued antimicrobial therapy when they we transferred to the ward from the ICU.

8. It is recommended to write the names of bacteria in accordance with medical spelling rules.

We have corrected the spelling of bacterial names (Throughout the manuscript and Table 2).

 

Reviewer #5: Järvisalo et al present a study of 126 patients with cultures positive sepsis and AKI and examined predictors of 90 day mortality. They found that elevated lactate level at admission and initiation of CRRT were associated with higher mortality level. The study has several shortcomings.

Major points:

1- The main problem I have with the study and the conclusion is I don't think the authors can truly say that this is the first paper that showed that elevated lactate was associated with higher mortality. I think this has been shown in numerous studies before.

The reviewer is right that previous data are available showing that lactate is associated with mortality in general patients with sepsis or septic shock according to Sepsis-3 criteria. However, to our knowledge this finding has not been previously observed in patients with blood culture positive septicemia and CRRT dependent AKI. To increase the novelty of our revised manuscript we have also included data on lactate clearance during the first 48 hours CRRT (Results: page 9, paragraph 2, lines 7-13; Figure 4; Discussion: page 10, paragraph 2, lines 24-27 and page 11, paragraph 1, lines 1-8). Furthermore, we have revised the text accordingly (Discussion: page 9, paragraph 4, lines 19-23; page 10, paragraph 1, lines 2-5 and paragraph 2, lines 15-17)

2- I have several reservation about the statistical analysis. The authors selected patients with sepsis based on the Sep3 definition which takes into account SOFA score. Than the authors adds two more severity of illness score (APACHE and SAPS) in the analysis. First, the is the problem of over-fitting when using so many severity of illness score. Second, since SOFA score was part of the SEP3 definition and therefore inclusion in the study, I am not sure it should be included in the multivariate. analyses.

The reviewer is right that using several ICU scoring systems might lead to overfitting in some cases. However, we examined multicollinearity by calculating variance inflation factors throughout the statistical analyses. We have revised the Statistical methods section of the manuscript accordingly (Methods: page 5, paragraph 2, lines 22-24). 

Furthermore, the multivariable models used were stepwise models with only explanatory variables with a significance P≤0.15 included in the final models to avoid overfitting. The final models did not include all SOFA, APACHE and SAPS but the most significant of these scores for each model, namely SAPS for the ICU admission model and APACHE for the CRRT initiation model. SOFA was, therefore, actually not included in the final multivariable models. Statistics consultation was available for the authors and employed for some of the analyses. 

3- I am having a hard time understanding the rationale for choosing lactate of 2 and 4 in the analysis. The authors should explain why those were chosen: was it based on the ROC? was is based on the univariate analyses? was is based on prior studies?

A lactate of 2 mmol/l was chosen as it is the upper normal level of lactate at our center. As for lactate exceeding 4 mmol/l (for the high lactate subgroup) was based on the receiver operating curve analysis (ROC AUC) data as a lactate of 4.3 mmol/l had a similar Youden's J index as a lactate of 5 mmol/l and the best sensitivity and specificity combination. Lactate of 4.3 mmol/l yielded a sensitivity of 0.74 and a corresponding specificity of 0.84 for discriminating patients deceased in the ICU from ICU survivors. The corresponding values for a lactate of 5 mol/l would have been sensitivity of 0.68 and specificity of 0.90. For practical issues we chose lactate >4 mmol/l and not lactate >4.3. We have included this point to the revised manuscript (Results: page 8, paragraph 5, lines 24-26 and page 9, paragraph 1, line 1).

 I am not sure this study adds much to the literature beyond what we already know about lactate levels in critically ill patients.

We have included data on lactate clearance during the first 48h of CRRT to increase the novelty and impact of our manuscript (Results: page 9, paragraph 2, lines 7-13; Figure 4; Discussion: page 10, paragraph 2, lines 24-27 and page 11, paragraph 1, lines 1-8).

---

## [Editor Report · Decision Letter 1]

22 Mar 2021

Mortality and Associated Risk Factors in Patients with Blood Culture Positive Sepsis and Acute Kidney Injury Requiring Continuous Renal Replacement Therapy - A Retrospective Study.

PONE-D-21-00589R1

Dear Dr. Järvisalo,

We’re pleased to inform you that your manuscript has been judged scientifically suitable for publication and will be formally accepted for publication once it meets all outstanding technical requirements.

Kind regards,

Aleksandar R. Zivkovic

Academic Editor

PLOS ONE

---

## [Editor Report · Acceptance letter]

23 Mar 2021

PONE-D-21-00589R1 

Mortality and Associated Risk Factors in Patients with Blood Culture Positive Sepsis and Acute Kidney Injury Requiring Continuous Renal Replacement Therapy - A Retrospective Study. 

Dear Dr. Järvisalo:

I'm pleased to inform you that your manuscript has been deemed suitable for publication in PLOS ONE. Congratulations! Your manuscript is now with our production department. 

Kind regards, 

on behalf of

Dr. Aleksandar R. Zivkovic 

Academic Editor

PLOS ONE